# Chemical mimicry of viral capsid self-assembly via corannulene-based pentatopic tectons

Yu-Sheng Chen[1], Ephrath Solel [2], Yi-Fan Huang[3], Chien-Lung Wang[3], Tsung-Han Tu [1], Ehud Keinan[2] & Yi-Tsu Chan [1]

Self-assembly of twelve pentatopic tectons, which have complementary edges or can be linked using either digonal or trigonal connectors, represents the optimal synthetic strategy to achieve spherical objects, such as chemical capsids. This process requires conditions that secure uninterrupted equilibria of binding and self-correction *en route* to the global energy minimum. Here we report the synthesis of a highly soluble, deca-heterosubstituted corannulene that bears five terpyridine ligands. Spontaneous self-assembly of twelve such tectons with 30 cadmium(II) cations produces a giant icosahedral capsid as a thermodynamically stable single product in high yield. Nuclear magnetic resonance (NMR) methods, mass spectrometry analyses, small-angle X-ray scattering, transmission electron microscopy, and atomic force microscopy indicate that this spherical capsid has an external diameter of nearly 6 nm and shell thickness of 1 nm, in agreement with molecular modeling. NMR and liquid chromatography evidences imply that chiral self-sorting complexation generates a racemic mixture of homochiral capsids.

[1] Department of Chemistry, National Taiwan University, Taipei 10617, Taiwan. [2] The Schulich Faculty of Chemistry, Technion-Israel Institute of Technology, Technion city, 32000 Haifa, Israel. [3] Department of Applied Chemistry, National Chiao Tung University, Hsinchu 30010, Taiwan. Correspondence and requests for materials should be addressed to E.K. (email: keinan@technion.ac.il) or to Y.-T.C. (email: ytchan@ntu.edu.tw)

Nature uses spherical containers ubiquitously in both the inanimate and living worlds, and the convex regular icosahedron (or its dual, dodecahedron) is the polyhedron that is closest in symmetry to the sphere. Consequently, the optimal bottom-up strategy to achieve spherical objects involves the construction of these Platonic solids. This notion has been applied at any scale, from dodecahedrane, Buckminsterfullerene, spherical virus capsids, as well as soccer balls and geodesic spheres. Spherical viruses use this approach to create particles ranging in size from 15 to 500 nm, which assemble spontaneously from their components under the proper conditions, and disassemble under other conditions, thus enabling the viral life cycle[1]. The icosahedral virus capsids teach us important lessons, including the economy of surface area-to-volume ratio and the genetic efficiency of subunit-based symmetric assembly.

Chemical mimicry of viral capsids is highly desirable because stable structures of icosahedral symmetry can be applied in many ways, including microencapsulation and transport of sensitive cargo, drug delivery and targeting, synthesis of nanoparticles of uniform size, reactivity modulation of bound guests, molecular recognition, catalysis, formation of structural elements for supramolecular architecture, and even safe immunization by epitope presentation on the surface of non-viral spherical objects.

We have previously proposed a general synthetic strategy for producing chemical capsids by self-assembly of 12 pentagonal tectons that have complementary edges or can be stitched together using either digonal or trigonal connectors (Fig. 1)[2]. We considered three different binding mechanisms: hydrogen bonding, metal binding, and formation of a dynamic covalent bonding, such as disulfide bonds[3,4]. As is the case with virus capsids, these structures are designed to assemble and disassemble under controlled environmental conditions.

Aiming at the total synthesis of the appropriate pentatopic subunits, we have considered the rigid corannulene skeleton, nicknamed buckybowl, which has a fivefold symmetry and suitable curvature[2]. We have realized that various *sym*-pentasubstituted corannulene derivatives could provide the appropriate positioning of desired functional groups around the rim of the bowl. Fortunately, the well-developed synthetic methodology for unidirectional substitution of corannulene, which is based on the ready availability of *sym*-pentachlorocorannulene[5], allows for the synthesis of such pentagonal tectons with the appropriate symmetry[6].

Although the concept seems straightforward, its experimental realization is non-trivial because the successful synthesis of bowl-shaped pentatopic tectons, addresses only part of the problem. Other major challenges include the requirement for the capsid thermodynamic stability, the need for kinetic flexibility of all intermediates[3], and the necessity for high solubility of all starting materials and intermediates in order to maintain a homogeneous mixture throughout the entire self-assembly process. The latter requirement is of crucial importance because the various

components can assemble in many alternative ways to produce off-pathway aggregates. Although the desired icosahedral capsid represents the global free energy minimum of the system, kinetic stability or insolubility of intermediates, would prevent the system from reaching the global minimum. Therefore, the entire system must be kept under conditions of uninterrupted fast equilibrium, allowing for self-correction.

Several attempts to achieve synthetic spherical capsids by self-assembly of 12 corannulene-based tectons bearing five functional groups have been reported. For example, Stang, Siegel and Baldridge have prepared corannulene derivatives with five ethynyl-platinum units[7], boronic esters[8], or single-stranded, self-complementary DNA chains[9]. Although these approaches seem highly appropriate, they probably produced various intermediates that were kinetically too stable to proceed to icosahedral capsids, which represent the global energy minimum. De Mendoza has achieved the closest realization of the pentagonal tectons strategy using 12 calix[5]arene pentacarboxylate units and 20 $UO_2^{2+}$ trigonal connectors[10]. However, the use of conical-shaped pentatopic units resulted in a stellated dodecahedron with concave faces on its exterior rather than a regular sphere.

Stang's approach to co-assemble tritopic pyridine ligands with linear Pt connectors, represents a remarkable advance on the quest to synthetic icosahedral objects[11,12]. The self-assembly of 20 triangles seems an easier entry to icosahedral objects, rather than the assembly of 12 pentagons, certainly from a synthetic standpoint. Nature, however, appears to prefer the latter strategy, probably because it is more versatile in terms of subunit diversity and function. More importantly, the dodecahedron is the single option available for pentagonal tectons to self-assemble into a low-energy closed object, whereas triangular tiles can assemble in multiple ways, including a tetrahedron, octahedron, icosahedron, and even a plane, as has been demonstrated by Nitschke[13].

Here we report the successful realization of our proposed strategy[2] to achieve spherical capsids by self-assembly of 12 shallow bowl-shaped pentatopic, corannulene-based tectons. A giant $[Cd_{30}L_{12}]$ icosahedral metallo-cage with an external diameter of ~6 nm and shell thickness of nearly 1 nm is self-assembled from deca-heterosubstituted corannulene derivative equipped with 2,2′:6′,2″-terpyridine (tpy) ligands. We confirm the structure of this coordination-driven chemical capsid using NMR methods, MS analyses, SAXS, TEM, AFM, and molecular modeling.

## Results

**Design and synthesis of pentatopic tectons.** Our initial efforts focused on the assembling of sulfur-containing corannulene tectons, such as 1,3,5,7,9-penta(*t*-butylsulfanyl)corannulene and 1,3,5,7,9-pentamercaptocorannulene, through metal-ligand coordination reactions or oxidation (Supplementary Fig. 1). Although our chemical strategy was strongly supported by molecular modeling, solubility problems obstructed proper purification and characterization of the desired capsids (Supplementary Figs 2–5).

Accordingly, we focused our efforts on achieving pentatopic tectons with much higher solubility. Thus, we switched to deca-heterosubstituted corannulene precursors, such as 1,3,5,7,9-pentamethoxy-2,4,6,8,10-pentabromocorannulene, **1** (Fig. 2), which was prepared by our previously described Cu(I)-catalyzed Ullmann condensation reaction between MeOH and *sym*-pentachlorocorannulene followed by electrophilic bromination[14]. The five methyl ethers in **1** increased solubility whereas the bromide groups served as handles for further cross-coupling to appropriate metal-binding functionalities.

Another significant change in our tecton design related to the metal-binding strategy. We decided to switch from the

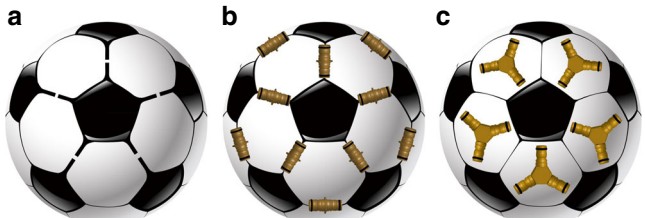

**Fig. 1** Three strategies for capsid formation from pentatopic tectons using the common image of a soccer ball: **a** Self-adhering tectons (black objects). **b** Tectons (black) with added digonal connectors (golden objects). **c** Tectons with added trigonal connectors (golden objects)

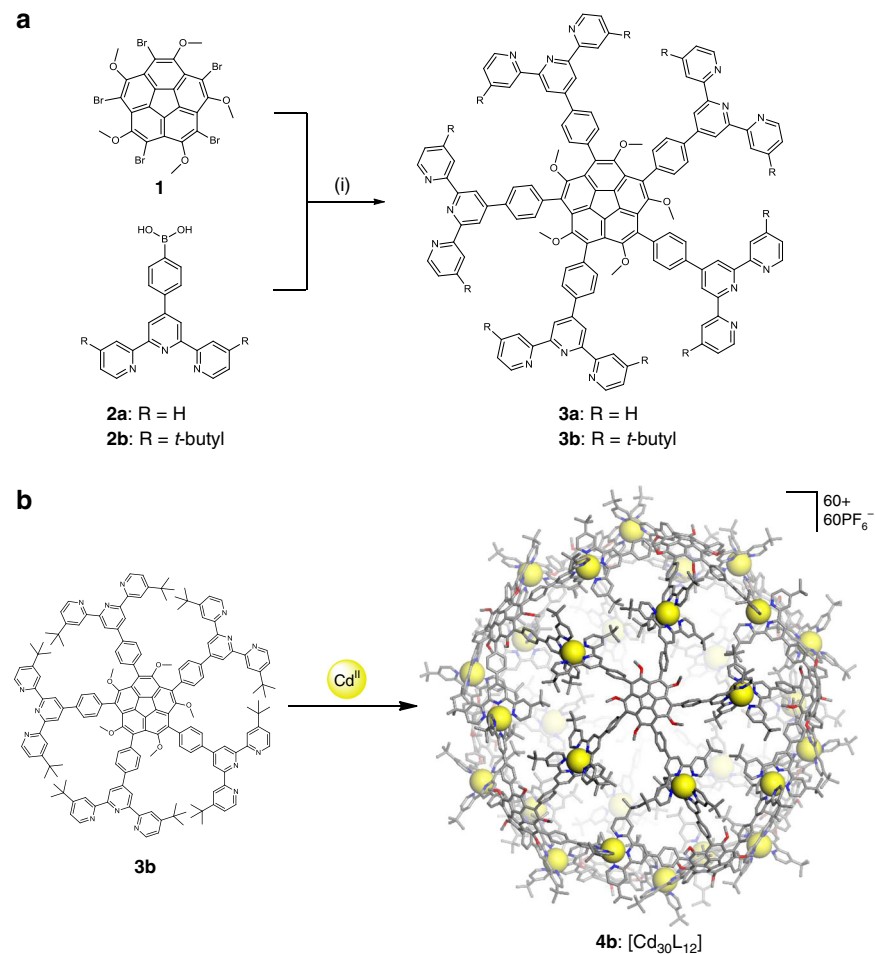

**Fig. 2** Ligand synthesis and molecular self-assembly. **a** Reagents and conditions for the synthesis of the corannulene tectons: (i) Na$_2$CO$_3$, Pd(PPh$_3$)$_4$, toluene/H$_2$O/*t*-BuOH (3/3/1, v/v/v), reflux. **b** Self-assembly of Cd(II) cations with pentatopic tectons, **3b**, to produce icosahedron [Cd$_{30}$L$_{12}$], **4b**. Supplementary Movie 1 shows the geometry-optimized capsid **4b**

monodentate sulfur-based ligands to the tridentate tpy function because this ligand offers high coordinative rigidity, enhanced solubility and propensity to bind most transition metals[15]. Tpy ligands are particularly advantageous for our goal because they bind many transition metal ions in a 2:1 molar ratio to give pseudo-octahedral [M(tpy)$_2$] complexes, offering a broad range of predictable thermodynamic stability: Cd(II) < Zn(II) < Fe(II) < Ru (II)[16]. This stability order allows for choosing the proper metal ion with the desired combination of thermodynamic stability and kinetic flexibility that would eventually lead to the desired capsid. Furthermore, the [M(tpy)$_2$] complexes are sufficiently stable to survive a broad range of pH values and temperatures, as well as the high electric field in ESI-MS experiments. For our design, the linearity of the <tpy-M-tpy> connection was crucial to secure the desired geometry for self-assembly, as had already been demonstrated in various Zn(II)- and Cd(II)-based polyhedral cage structures[17].

Hence, we synthesized tpy-containing phenylboronic acid derivatives, **2a** (R = H) and **2b** (R = *t*-butyl) according to the reported procedure of Wang et al.[18]. The *t*-butyl groups at the terpyridyl 4,4″-positions were intentionally introduced to further enhance the solubility in organic solvents. Finally, the Suzuki-Miyaura cross-coupling reaction[19] of **1** with either **2a** or **2b** afforded either 1,3,5,7,9-pentamethoxy-2,4,6,8,10-penta[4-(4′-terpyridyl)phenyl]corannulene, **3a**, or 1,3,5,7,9-pentamethoxy-2,4,6,8,10-penta[4-(4′-4,4″-di-*t*-butylterpyridyl)phenyl]corannulene,

**3b**, in 49% or 57% yield, respectively (Fig. 2a). Considering the fact that the reaction involved five coupling sites on the same molecule, these yields reflect nearly 90% yield per site. These pentatopic ligands were purified by column chromatography using amino-coated silica gel. We based the structural characterization on NMR spectroscopy and MALDI-TOF-MS analysis (Supplementary Figs 6–15).

**Self-assembly and characterization of capsid 4b.** Since the complexation reaction of **3a** with Cd(II) cations gave rise to metal complexes having poor solubility in common organic solvents, we switched to the *t*-butyl-substituted analog, **3b**. Thus, we mixed a CHCl$_3$ solution of **3b** (1 equiv) with a MeOH solution of Cd (NO$_3$)$_2$·4H$_2$O (2.5 equiv) at room temperature. Addition of excess NH$_4$PF$_6$ (30 equiv with respect to NO$_3^-$) resulted in precipitation of the corresponding PF$_6^-$ salt. The latter was re-dissolved in acetonitrile and the solution was further refluxed for 12 h to afford [Cd$_{30}$L$_{12}$], **4b**, in high yield (88%).

The $^1$H NMR spectrum of **4b** (Fig. 3a) exhibits a single set of clear signals that can be easily assigned to specific protons. In comparison with the free ligand, **3b**, the most significant change upon metal complexation is an upfield shift of 6,6″-tpy protons, which strongly supports the pseudo-octahedral coordination geometry of the <tpy-Cd(II)-tpy> complex[20]. The sharp resonance at δ = 9.02 ppm, assigned to the 3′,5′-tpy protons,

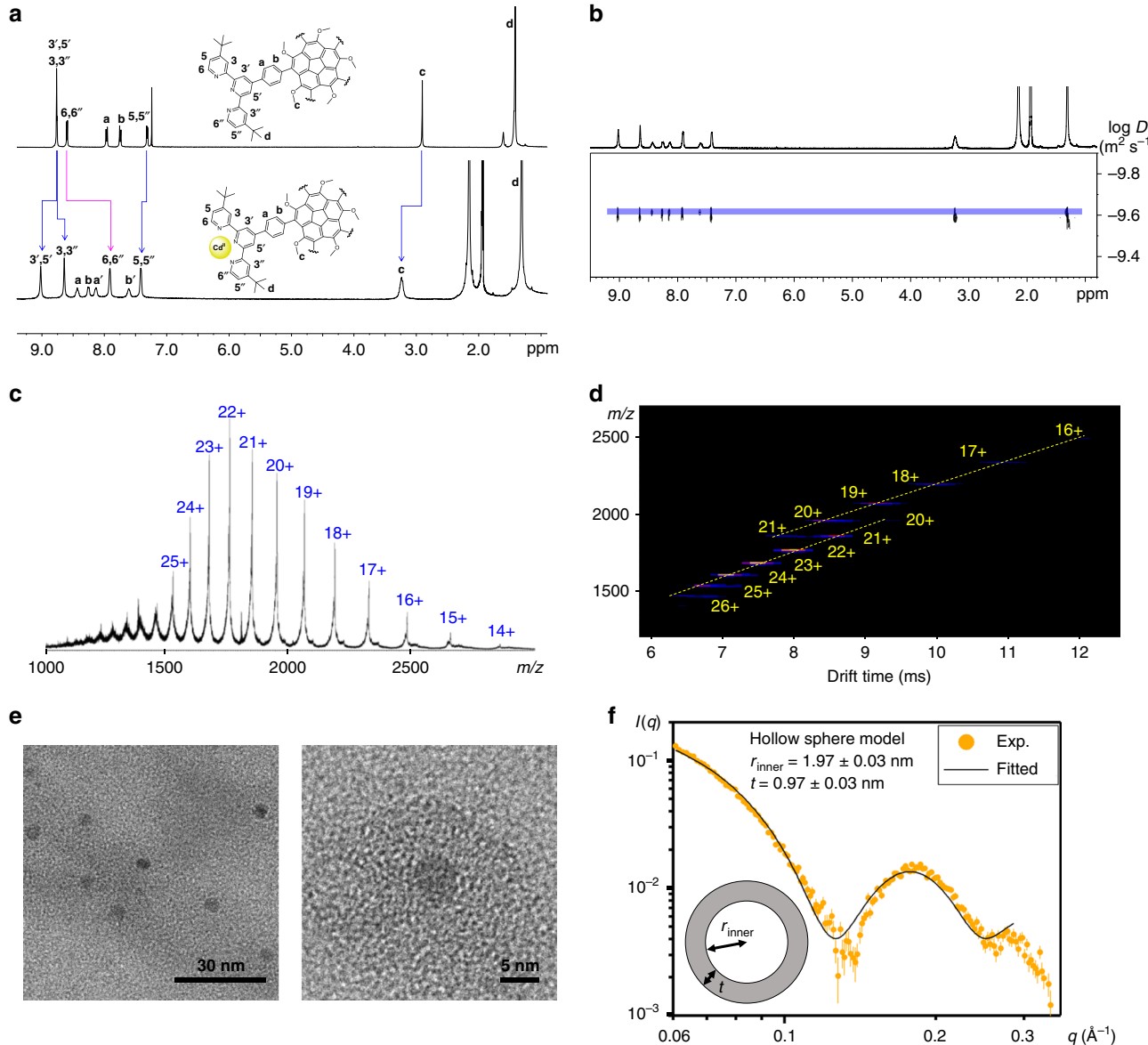

**Fig. 3** Structural characterization of self-assembled **4b**. **a** $^1$H NMR spectra of ligand **3b** in CDCl$_3$ (top) and **4b** in CD$_3$CN (bottom). **b** DOSY NMR spectrum of **4b** in CD$_3$CN at 25 °C. **c** ESI-MS spectrum. **d** ESI-TWIM-MS plot. **e** TEM micrographs of the samples prepared by drop casting a dilute solution of **4b** ($10^{-6}$–$10^{-7}$ M) in acetonitrile onto a carbon-coated copper grid. **f** SAXS profile of **4b** (0.5 mg mL$^{-1}$ in acetonitrile) and the fitting result derived from the hollow sphere model. The error bars represent the uncertainties in the scattering intensity

suggests that these protons reside in the same average environment, indicating fast rotation of the pseudo-octahedral unit around two single bonds. The 2D COSY and ROESY/NOESY NMR spectra (Supplementary Figs 17, 18) exhibit four minor peaks assigned to protons a, a′, b, and b′ on the phenylene ring. This observed peak splitting, which occurs upon complexation, reflects slow rotation of the phenylene ring around the single bond connecting it to the corannulene moiety. The EXSY experiments further verify that the peak splitting is involved in a slow exchange with a rate constant of 3.64 s$^{-1}$ at 25 °C (Supplementary Fig. 19). This conclusion is also supported by variable temperature NMR experiments (Supplementary Fig. 20), which exhibit line broadening of the phenylene protons upon warming from 298 to 343 K, whereas the tpy signals do not vary much with temperature. Consequently, the fast rotation of the < tpy-Cd(II)-tpy > unit occurs around the single bonds connecting the phenylene ring to the tpy moiety.

The DOSY experiment (Fig. 3b) shows that all signals share the same diffusion coefficient ($D = 2.40 \pm 0.06 \times 10^{-10}$ m$^2$ s$^{-1}$) at 25 °C in CD$_3$CN, confirming the existence of a single species. The corresponding hydrodynamic radius ($r_H$) calculated by the Stokes-Einstein equation is 2.48 ± 0.05 nm, which agrees with the dimension of the energy-minimized capsid structure (Supplementary Table 2). Moreover, the $^{113}$Cd NMR spectrum (Supplementary Fig. 22) features a single peak at 275.64 ppm, indicating that all Cd(II) centers reside in an identical chemical environment. Notably, the sharp $^{17}$F and $^{31}$P NMR resonances (Supplementary Figs 23, 24) of the PF$_6^-$ ions imply that the counterions can freely move in and out of the capsid on the NMR time scale.

The bowl shape of the corannulene core and the directional substitution around it render the free ligand, **3b**, a chiral molecule, although the rapid bowl-to-bowl inversion at room temperature efficiently interconverts the two enantiomers[21].

Nevertheless, complexation to the metal ions locks the individual ligands at one enantiomeric form. The single set of terpyridyl signals observed in the $^1$H NMR spectrum implies that a chiral self-sorting event occurs upon complexation, which generates a racemic mixture of homochiral capsids. The chiral self-sorting is also evident from the silent circular dichroism (CD) spectrum (Supplementary Fig. 25b). Nevertheless this racemate was resolved by high-performance liquid chromatography (HPLC) using a chiral column, revealing two distinct peaks with identical intensities (Supplementary Fig. 26).

ESI-MS experiments confirmed the expected composition of $[Cd_{30}L_{12}]$ (Fig. 3c), exhibiting a series of signals derived from the 14+ to 25+ ions, which correspond to $\{[Cd_{30}L_{12}]-nPF_6\}^{n+}$ (where $n = 14\sim25$). Although the isotope pattern of each charge state was unclear in the mass measurements, the consecutive signals matched perfectly with the simulated MS of the icosahedral complex, **4b**. The unclear isotope patterns could result from the high molecular weight (~42 kDa) and high charge states. Traveling wave ion mobility mass spectrometry (TWIM-MS)[22] provided further structural information. The ESI-TWIM-MS spectrum of **4b** (Fig. 3d) showed two signal distributions ranging from 16 + to 26 + charge states, which presumably result from the enhanced Coulombic repulsion at the higher charge states, leading to inflated conformations[23]. The experimental collision cross-sections (CCSs) were calculated according to the drift times measured by ESI-TWIM-MS. The average experimental CCS agreed well with the CCSs derived from the energy-minimized structures, further supporting the formation of the expected icosahedral capsid (Supplementary Table 1).

The morphology of the individual capsid was examined by TEM (Fig. 3e) with samples prepared by drop casting of dilute acetonitrile solutions ($10^{-6}\sim10^{-7}$ M) onto copper grids. All cage molecules were well distributed on a copper grid, exhibiting an average diameter of $4.90 \pm 0.42$ nm (Supplementary Fig. 27), which is consistent with the simulated value. Moreover, the AFM image of **4b** on mica (Supplementary Fig. 28) displayed an average height of $5.80 \pm 0.48$ nm, which agrees with the modeled external diameter of $6.02 \pm 0.14$ nm (Supplementary Table 2).

The detailed morphology of **4b** in acetonitrile was further studied by small-angle X-ray scattering (SAXS). The mesoscopic shape and size of the capsid can be obtained by fitting the experimental SAXS profile to a specific topological model[24]. The SAXS profile of **4b** (Fig. 3f) cannot be fitted by the density function of a monodisperse hard sphere model, but can be well-fitted by the density function of a monodisperse hollow sphere model (black line in Fig. 3f). The well-fitted result confirmed that the 12 bowl-shaped corannulene-based tectons indeed self-assemble into a hollow sphere structure with an inner radius ($r_{inner}$) of $1.97 \pm 0.03$ nm and a shell thickness ($t$) of $0.97 \pm 0.03$ nm. The shell thickness coincides with the distance between the two $t$-butyl groups on positions 4 and 4″ on the tpy unit. The SAXS mean radius ($r_{mean}$) of the hollow sphere ($2.46 \pm 0.03$ nm) is consistent with the dimension of the simulated structure (Supplementary Fig. 29 and Table 2), thus confirming the simulated hollow sphere structure and dimension of **4b**.

## Discussion

Icosahedral and dodecahedral assemblies are the pinnacle of Platonic solid constructions. Synthesis of such entities represents the optimal bottom-up strategy to achieve spherical objects in both the inanimate and living worlds. Twelve years ago, inspired by the spherical virus capsids, we have proposed a general synthetic strategy for producing chemical capsids by self-assembly of 12 bowl-shaped, pentatopic, corannulene-based tectons[2]. Although this goal has long been a Holy Grail for the supramolecular community, most of the previous synthetic attempts were hampered by either solubility problems or high kinetic stability of off-track aggregates. These problems also interfered with our own attempts to create capsids from either pentamercapto-corannulene or the corresponding penta-thioethers utilizing reversible reactions, such as formation of disulfide bonds or metal complexation (Supplementary Fig. 30).

Switching to deca-heterosubstituted corannulene derivatives equipped with five ether functions and five tpy ligands, such as **3b**, afforded the desired solubility and coordinative rigidity. Tpy ligands were particularly advantageous as linear connectors because they can form pseudo-octahedral [M(tpy)₂] complexes with many metals, and thereby offer a delicate balance between thermodynamic stability and kinetic flexibility. Cadmium(II), which forms the weakest and therefore the kinetically most flexible metal complex with these ligands, was found to be the best connector. Thus, a giant icosahedral metallo-cage was eventually self-assembled from 12 corannulene-based tectons and 30 Cd(II) linkers.

Accurate structural information of these chemical capsids was obtained by NMR methods, MS analyses, SAXS, TEM, AFM, and molecular modeling. These methods provided not only the capsid external diameter of nearly 6 nm and shell thickness of 1 nm, but also the purity level and dynamic properties. Furthermore, chiral self-sorting phenomena during the assembly process have resulted in the formation of a racemic mixture of homochiral capsids.

The substantial size and molecular weight of these chemical capsids, as well as their ready availability and chemical stability, offer a plethora of potential applications, including micro-encapsulation and transport of sensitive cargo, drug delivery and targeting, synthesis of monodisperse nanoparticles, reactivity modulation of bound guests, molecular recognition, asymmetric synthesis, catalysis, safe immunization and more. Many of these opportunities are currently being explored in our laboratories.

Finally, those who wonder why it took so long to achieve this seemingly simple realization of chemical capsids may recall the words of Pierre-Auguste Renoir, "This drawing took me five minutes, but it took me sixty years to get there."

## Data availability
The authors declare that the data supporting the findings of this study are available within the Supplementary Information files and from the corresponding authors upon reasonable request.

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

## Acknowledgements

This research was supported by the Ministry of Science and Technology of Taiwan (MOST106-2628-M-002-007-MY3). The authors thank the National Synchrotron Radiation Research Center (NSRRC) in Taiwan for assistance with the SAXS measurements.

## Author contributions

E.K. and Y.T.C. conceived the project. Y.S.C. and E.S. performed the synthesis and characterization experiments. Y.F.H., C.L.W., and T.H.T. performed the SAXS measurements and simulations. T.H.T. performed the TEM and AFM measurements. E.K. and Y.T.C. analyzed the data and wrote the paper. All authors discussed the results and commented on the paper.

## Additional information

**Competing interests:** The authors declare no competing interests.

