## [Peer Review File · Nature Communications]

Reviewers' Comments:

Reviewer #1:

Remarks to the Author:

The paper by Chan et. al reported the self-assembly of a giant icosahedral cage using five-armed terpyridine building block. In the field of coordination driven self-assembly, tetrahedron, cube, octahedron and dodecahedron were reported in both pyridine and terpyridine-based system. Icosahedron is the most challenging one left in this field. So this report will attract broad readership in the community. The following issues needs to be addressed specifically before publication.

1 .The author gave the modeling icosahedral structures for sulfur-based ligands. However, most of these ligands lack sufficient directionality for the formation of proposed structures. The sulfur-based ligands may more prefer forming dimer-like structures rather forming large icosahedral. Furthermore, the authors didn't give any solid evidence for the formation of the proposed structure. Any aggregation form such as disordered clusters could give the element signals showing in EDS or EELS. If no further evidences were obtained, the authors should consider removing them from the manuscript or discussing the results in supporting information.

2. The NMR data of the a,b proton on ligand 6b split into four sets of peaks when self-assembled into 7b, and the author claimed that it came from the rotatory inhibitions of the phenyl ring. If this is the case, the rotatory inhibition should also be observed in ligand 6b because of the methoxy groups, why did NMR spectra only show two sets of peaks? VT-NMR of 7b showed line broadening, what about the ligand? 1D NOE experiments are also needed to verify the conformation of the phenyl ring. Moreover, some NMR data were missing, e.g., ¹³C spectra for supramolecule 7b; 2D NMR (COSY, ROESY/NOESY) of ligand 6a, 6b; COSY, aliphatic region of NOESY/ROESY spectra of 7b, especially the spatial correlation between proton b and c.

3. If single crystal was not obtained, the authors should consider AFM characterization of the height and size of the supramolecule.

4. The self-assembly process was claimed to be a chiral self-sorting process, but neither experimental evidence nor theoretical calculation was provided. The authors mentioned the ligand is actually a chiral compound. Two enantiomers both exist. However, there is no evidence to prove that these two enantiomers undergo self-sorting process during complexation. Are some ligands in the supramolecule the other enantiomer based on NMR results? Or there could be enantiomers formed during the process, which may also lead to the splitting of the a, b signal on the phenyl ring. Is that possible to separate these two enantiomers first and then test the self-sorting behavior by CD?

Reviewer #2:

This manuscript is a substantial contribution to the area of polyhedral supramolecular assemblies. Icosahedral and dodecahedral assemblies are the pinnacle of Platonic solid constructions. The 5-fold symmetry is special in biology and exceptional in 2-D lattices (only possible in quasi-crystals). Nature offers the viral capsid, which protein chemists have managed to imitate; however ready assemblies from 5-fold symmetric building block like corannulene have long been a Holy Grail for the supramolecular community.

The present manuscript offers strong evidence that such supramolecular assemblies can form from 5-fold symmetric corannulenes, substituted with metal binding ligands like aryl terpyridines. Through a combination mass spectrometric and microscopy methods it is reasonable to assume that dodcahedral

Platonic molecular capsids form by assembly of the designed corannulene-base tectons combined with Cd metal ions.

The size and molecular weight of these objects is substantial, which is important here because the symmetry makes it difficult to distinguish monomers from dodecamers by standard NMR techniques. The authors made clever use of relaxation time measurements that depend on the tumbling/migrating rate of the molecules, which inherently depends on the size and molecular weight.

As with many great milestone discoveries in science, once people see this it may seem "easy". One can only hope that a group of younger scientists will think just that and attack this problem as if it were low hanging but recently ripened fruit. Then all of us will enjoy the many creative new structures, properties and uses these chemical capsids are bound to offer.

I highly recommend publishing this article.

Jay Siegel

Reviewer #3:

Remarks to the Author:

In this manuscript the authors describe the development (with failures and success) of corannulene based ligands for the self-assembly of a sphere like supramolecular structure. While sulfur based ligands were not successful, final success has been reached with terpyridines. Since the work of Stang or Fujita, big spheres are well established in the field of supramolecular chemistry. Here another but spectacular example following the reported principles is described. The paper reads like a progress report giving all information and not only the scientifically important ones. I do not feel that this paper is appropriate for Nature Communication. I rather would recommend to publish it in a more chemically orientated journal (JACS, ACIE).

May 20, 2019

Reviewer #1:

Comment 1 –

The paper by Chan et. al reported the self-assembly of a giant icosahedral cage using five-armed terpyridine building block. In the field of coordination driven self-assembly, tetrahedron, cube, octahedron and dodecahedron were reported in both pyridine and terpyridine-based system. Icosahedron is the most challenging one left in this field. So this report will attract broad readership in the community.

⇒ We highly appreciate these encouraging remarks.

Comment 2 –

The author gave the modeling icosahedral structures for sulfur-based ligands. However, most of these ligands lack sufficient directionality for the formation of proposed structures. The sulfur-based ligands may more prefer forming dimer-like structures rather forming large icosahedral. Furthermore, the authors didn't give any solid evidence for the formation of the proposed structure. Any aggregation form such as disordered clusters could give the element signals showing in EDS or EELS. If no further evidences were obtained, the authors should consider removing them from the manuscript or discussing the results in supporting information.

⇒ As suggested, the sulfur-based results have been moved to the Supplementary Information.

Comment 3 –

The NMR data of the a,b proton on ligand 6b split into four sets of peaks when self-assembled into 7b, and the author claimed that it came from the rotatory inhibitions of the phenyl ring. If this is the case, the rotatory inhibition should also be observed in ligand 6b

because of the methoxy groups, why did NMR spectra only show two sets of peaks? VT-NMR of **7b** showed line broadening, what about the ligand? 1D NOE experiments are also needed to verify the conformation of the phenyl ring.

⇒ As suggested, we conducted the variable-temperature NMR study on ligand **3b** (**6b** in the original manuscript). Even at 223 K, no peak splitting for the phenylene ring of **3b** was observed. This result could be attributed to the fast bowl-to-bowl corannulene inversion, which made the phenylene protons have the same chemical environment in the uncomplexed ligand. However, the bowl-to-bowl inversion is restricted after complexation. Consequently, the peak splitting for protons a and b was observed. The VT NMR spectra of **3b** is now included in the Supplementary Information (Fig. 16).

⇒ To further confirm the slow rotation of the phenylene unit in the capsid, an EXSY experiment was conducted. The ^1H - ^1H 2D EXSY spectrum (see below) revealed the exchange cross peaks (red) between the two signals for both a,a' and b,b', strongly indicating that the two protons are involved in a slow exchange. This observation also excluded the possibility that the peak splitting is derived from the formation of diastereomers. Moreover, the exchange rate constant can be determined by Equations 1 and 2 where k is the exchange rate constant and T_m is the mixing time used in the EXSY measurement. Using signals b and b' as an example, I_{bb} and $I_{b'b'}$ are the volumes of diagonal peaks and $I_{bb'}$ and $I_{b'b}$ are those of cross peaks. With varying mixing times, the exchange rate constant was estimated to be 3.64 s^{-1} at $25 \text{ }^\circ\text{C}$ based on the plot shown below. The above discussion and EXSY results have been added to the Supplementary Information, and the statement of “The EXSY experiments further verify the peak splitting is involved in a slow exchange with a rate constant of 3.64 s^{-1} at $25 \text{ }^\circ\text{C}$ (Supplementary Fig. 19).” has been included in the manuscript.

Comment 4 –

Moreover, some NMR data were missing, e.g., ^{13}C spectra for supramolecule **7b**; 2D NMR (COSY, ROESY/NOESY) of ligand **6a**, **6b**; COSY, aliphatic region of NOESY/ROESY spectra of **7b**, especially the spatial correlation between proton **b** and **c**.

⇒ The requested NMR data have been added to the Supplementary Information, including ^{13}C NMR of the capsid **7b** (now **4b**, Fig. 21), COSY and NOESY of ligand **6a** (now **3a**, Figs 7-8), COSY and NOESY of ligand **6b** (now **3b**, Figs 12-13), and COSY and NOESY of **4b** (Figs 17-18).

Comment 5 –

If single crystal was not obtained, the authors should consider AFM characterization of the height and size of the supramolecule.

⇒ The AFM characterization of **4b** on mica has been included in the Supplementary Information (Fig. 28). The AFM image revealed an average height of 5.80 ± 0.48 nm, which agrees well with the modeled external diameter of 6.02 ± 0.14 nm. However, the tip-sample convolution effect (*Science*, **1994**, 265, 1577-1579) resulted in the low accuracy of the measured diameter (41.40 ± 3.71 nm). The results have been included in the manuscript and Supplementary Information.

No.	Height (nm)	Diameter (nm)
1	6.73	42.65
2	5.98	39.32
3	6.21	39.55
4	5.86	45.98
5	5.66	41.94
6	5.34	47.39
7	5.34	40.98
8	5.88	39.79
9	5.23	35.02

Average Height: 5.80 ± 0.48 nm

Average Diameter: 41.40 ± 3.71 nm

Comment 6 –

The self-assembly process was claimed to be a chiral self-sorting process, but neither experimental evidence nor theoretical calculation was provided. The authors mentioned the ligand is actually a chiral compound. Two enantiomers both exist. However, there is no evidence to prove that these two enantiomers undergo self-sorting process during

complexation. Are some ligands in the supramolecule the other enantiomer based on NMR results? Or there could be enantiomers formed during the process, which may also lead to the splitting of the a, b signal on the phenyl ring. Is that possible to separate these two enantiomers first and then test the self-sorting behavior by CD?

⇒ The single set of terpyridyl signals observed in the ^1H NMR spectrum implies that a chiral self-sorting event occurs upon complexation, which generates a racemic mixture of homochiral capsids. The NMR evidence excluded the possibility of forming diastereomeric mixtures, which would be expected to exhibit intricate NMR spectra. As discussed in our response to Comment 3, the EXSY experiments indicate that the peak splitting reflects a slow exchange rather than formation of stereoisomers. The racemic mixture is evident from the silent CD spectrum of the mixture (see below). An enantiopure capsid cannot be prepared via asymmetric synthesis because the free ligand racemizes rapidly through the bowl-to-bowl inversion. Instead, we found that the capsids can be resolved to two homochiral enantiomeric capsids using chiral HPLC. The detailed experimental data have been included in the Supplementary Information and a short discussion has been added in the manuscript.

Reviewer #2:

Comment –

In This manuscript is a substantial contribution to the area of polyhedral supramolecular assemblies. Icosahedral and dodecahedral assemblies are the pinnacle of Platonic solid constructions. The 5-fold symmetry is special in biology and exceptional in 2-D lattices (only possible in quasi-crystals). Nature offers the viral capsid, which protein chemists have managed to imitate; however ready assemblies from 5-fold symmetric building block like corannulene have long been a Holy Grail for the supramolecular community.

The present manuscript offers strong evidence that such supramolecular assemblies can form from 5-fold symmetric corannulenes, substituted with metal binding ligands like

aryl terpyridines. Through a combination mass spectrometric and microscopy methods it is reasonable to assume that dodecahedral Platonic molecular capsids form by assembly of the designed corannulene-base tectons combined with Cd metal ions.

The size and molecular weight of these objects is substantial, which is important here because the symmetry makes it difficult to distinguish monomers from dodecamers by standard NMR techniques. The authors made clever use of relaxation time measurements that depend on the tumbling/migrating rate of the molecules, which inherently depends on the size and molecular weight.

As with many great milestone discoveries in science, once people see this it may seem "easy". One can only hope that a group of younger scientists will think just that and attack this problem as if it were low hanging but recently ripened fruit. Then all of us will enjoy the many creative new structures, properties and uses these chemical capsids are bound to offer.

I highly recommend publishing this article.

⇒ We highly appreciate these encouraging remarks and thank the reviewer for highlighting important aspects of our work. In fact, we have adopted few words from this text and included them in the conclusion part.

Reviewer #3:

Comment –

In this manuscript the authors describe the development (with failures and success) of corannulene based ligands for the self-assembly of a sphere like supramolecular structure. While sulfur based ligands were not successful, final success has been reached with terpyridines. Since the work of Stang or Fujita, big spheres are well established in the field of supramolecular chemistry. Here another but spectacular example following the reported principles is described. The paper reads like a progress report giving all information and not only the scientifically important ones. I do not feel that this paper is appropriate for Nature Communication. I rather would recommend to publish it in a more chemically orientated journal (JACS, ACIE).

⇒ We are encouraged by the statement of this reviewer that our work is of the high quality that is appropriate for the top tier chemistry journals, such as JACS or ACIE (and we think that Nature Chemistry should be included in this list). However, we disagree about the expected general impact of our report and its relevance to multiple fields beyond chemistry. Although our “failure” stories with the sulfur-based tiles were essential steps on our way to achieve the “success” story, we accept the reviewer’s opinion that those “failure” stories should be removed from the main text and be transferred to the Supplementary Information. In fact, this proposal overlaps with comment 2 of Reviewer 1, so we can respond in the same way: we moved the

entire sulfur-based results to the Supplementary Information. We still believe that these results are very important for other scientists who wish to follow up on our research and therefore should be accessible, but we agree that it is not essential to include them in the main text.

With many thanks and best regards,

Yi-Tsu Chan, Ph.D.
Department of Chemistry
National Taiwan University

Reviewers' Comments:

Reviewer #1:

Remarks to the Author:

The authors have addressed all the concerns and the manuscript is ready for publishing on Nature Communications.